# Investment Trade-Off between Mating Behavior and Tonic Immobility in the Sweetpotato Weevil *Cylas formicarius* (Coleoptera: Brentidae)

**DOI:** 10.3390/insects14010073

**Published:** 2023-01-12

**Authors:** Haoyong Ouyang, Runzhi Zhang, Muhammad Haseeb

**Affiliations:** 1State Key Laboratory of Integrated Management of Pest Insects and Rodents, Institute of Zoology, Chinese Academy of Sciences, Beijing 100101, China; 2College of Life Science, University of Chinese Academy of Sciences, No. 19(A) Yuquan Road, Shijingshan District, Beijing 100049, China; 3Center for Biological Control, College of Agriculture and Food Sciences, Florida A&M University, Tallahassee, FL 32307, USA

**Keywords:** anti-predator behavior, tonic immobility, mating behavior, *Cylas formicarius*

## Abstract

**Simple Summary:**

Tonic immobility (TI) is an essential anti-predator behavior to reduce the effect of predators. To know the relationship between mating behavior and TI, the effect of TI on courtship and copulation, as well as the effect of courtship and copulation on TI, were investigated in the sweetpotato weevil (SPW), *Cylas formicarius*. In the present study, we found the duration of TI was significantly reduced in the stage of courtship and copulation. The pairs with males from the L-strain (SPW with longer period of TI) showed lower frequency and longer duration of courtship than pairs with males from the S-strain (SPW with shorter duration of TI). Similarly, males from L-strain pairs showed a longer period of copulation than pairs with males from the S-strain. However, there is no significant difference in the frequency of copulation and the success of insemination. The results of our study clearly demonstrate that SPW mating behavior and TI are negatively correlated.

**Abstract:**

Numerous studies have confirmed that the trade-off between anti-predator behavior and mating behavior occurs in certain insect species. This suggests that insects invest more in anti-predator behavior, and fewer resources or time can be used in mating behavior. However, few studies focus on tonic immobility, an important anti-predator behavior in nature, and different stages in mating behavior. Tonic immobility (TI) is considered to be an important anti-predator behavior. Herein, we investigated the relationship between TI and mating behavior in the sweetpotato weevil (SPW), *Cylas formicarius*. As the first step, we artificially selected SPWs for the longer duration of TI (L-strain) and the shorter duration of TI (S-strain). The effect of courtship and copulation on the duration of TI in two artificial selection strains was tested. Furthermore, we compared the frequency and duration of two mating behaviors in four kinds of pairs (LF×LM, LF×SM, SF×LM, and SF×SM: LM—L-strain male; SM—S-strain male; LF—L-strain female; SF—S-strain female). Finally, we tested insemination success in four kinds of pairs (male and female SPWs from the L-strain or the S-strain). The courtship and copulation significantly reduced the duration of TI. Pairs with males from the L-strain showed lower frequency and longer duration of courtship than pairs with males from the S-strain. Similarly, males from L-strain pairs showed a longer period of copulation than pairs with males from the S-strain. However, there is no significant difference in the frequency of copulation and the success of insemination. These results support that there was a significant trade-off between TI and courtship as well as copulation in the SPW.

## 1. Introduction

Playing dead or feigning death is a behavior in which an insect shows no life. This form of insect deception is an adaptive behavior, also known as tonic immobility (TI) or thanatosis. In this approach, prey displays a distinctive posture that no longer responds to external stimuli when stimulated by predators or the environment. The presence of TI has been reported in a number of taxonomic studies for many years. In invertebrates, the presence of TI has been confirmed in spiders [1], butterflies [2], beetles [3,4,5], juvenile dragonflies [6], and ants [7]. In vertebrates, TI has been reported in birds [8,9], fishes, amphibians [10], and snakes [11,12].

Prey evolve a variety of physiological, behavioral, and morphological traits to reduce their predatory influences, such as combat, escape, and mimicry [13]. There are six stages of predation: two individuals in proximity, detection, identification, contacting, subduing, and consuming [14]. TI has been reported to be an effective anti-predator strategy in improving fitness during the subduing and consuming stages [15].

There are several phases to mating behavior, including mate searching, courtship, copulation, insemination, and post-copulation interactions [16]. Both courtship and copulation are important and costly stages of mating behavior in insects. Courtship is the close-range intersexual behavior which is considered to be a tool for sexual selection and specific-mate recognition for mating [17]. Copulation is one of the reproductive stages in which sperm transfer happens in internally fertilizing animals. There are several studies that reported that both stages are energetically taxing and resource-intensive behavior [17,18,19]. For example, due to the costly courtship behavior, male dung beetles *Onthophagus binodis* showed reduced longevity when housed with females [20]. Reproduction is often considered a trade-off against insect mobility [20].

A central premise in life history theory posits an evolutionary trade-off between survival and reproduction [17]. The trade-off between survival and reproduction is considered to be caused by the competitive allocation of limited resources into reproduction traits versus survival [21,22]. Most studies have shown that mating behavior is particularly likely to experience increased predation risk. Mating behavior increases the influence of predation, either by increasing conspicuousness, reducing mobility, or inhibiting the anti-predator behavior in prey. For example, by increasing investment in copulation, Australian plague locusts *Chortoicetes terminifera* raise their risk of parasitoid-mediated death by *Sphex cognatus* [23]. Thus, it is essential to investigate the effect of mating behavior on anti-predator behavior to find the reason for increasing predator risk. TI, a necessary anti-predator behavior, can be measured easily under the laboratory environment to quantify the anti-predator behavior. It is a good method to investigate the relationship between anti-predator and reproduction behavior.

The sweetpotato weevil (SPW), *Cylas formicarius* (Coleoptera: Brentidae), exhibits TI after stimulation [24]. *C. formicarius* has evolved TI as a defensive strategy to avoid predation. A predator or other external stimulus stimulates the SPW’s body to become motionless and to fall from its host plants. This causes the predator to have difficulty identifying its location. It has been reported that mites, ants, and mice prey on SPWs [24,25]. SPWs can be distinguished based on discrimination characteristics, such as the posture of their legs, antennae, and head [5] (Figure 1).

To further know the relationship between survival and reproduction, it is vital to assess the interaction between anti-predator behavior and mating behavior. Previous studies found that male and female SPWs reduced TI duration after copulation [26]. We confirmed that the trade-off between anti-predator behavior and mate searching exists in the male SPW [27]. Thus, we hypothesized that there is a trade-off between TI and courtship and copulation. To determine the relationship between courtship and copulation on TI in the present study, we measured whether the duration of TI of SPWs was effect by the stages of courtship and copulation. The frequency and duration of courtship and copulation of the SPW in two artificial strains with longer and shorter duration of TI were compared. In the end, after the copulation of two artificial strains of male and female SPWs, we detected whether the females were fertilized or not. 

## 2. Materials and Methods 

### 2.1. Insects Culture

The original SPW colony was obtained from the Tropical Research and Education Center, University of Florida, Homestead, FL, USA, in June 2018. Weevils were raised on sweet potato tubers (Bonnie Plants, Union Springs, AL, USA) under the following conditions: 25 ± 1 °C, 70–80% relative humidity, and a photoperiod of 12:12 h (light from 06:00 to 17:59). Eleven generations of SPW were raised at the Center for Biological Control, Florida A&M University, Tallahassee, FL, USA. The distal segments of the antenna were used to distinguish adult males from adult females of the SPW. We maintained newly emerged adult males and females in separate Petri dishes (15 cm diameter, 2 cm height) that contained 30 adults and 50 g of sweet potato tubers. Mass-reared virgin adult males and females of the SPW were used in this study.

### 2.2. Observations of TI Behavior

A 14–16-day-old adult SPW was collected in a plastic dish to observe the duration of TI. Before the observation, each weevil was weighed using an electronic balance (A&D ER-60A, Tokyo, Japan). To reduce the effect of same-sex sexual behavior and disturbance, each SPW used in the experiment was kept in separate plastic containers with 5 g of sweet potato tubers for 24 h after the weight measurement [28]. After isolation and treatment, TI was artificially induced in each weevil by using forceps (Bio Quip 4750, Rancho Dominguez, CA, USA) to grasp the abdomen and then drop it into its dish from a height of about 2 cm. The period between the dropping of a weevil into the dish and the detection of the first visible movement was recorded. If the weevil failed to respond to the first stimulation, we repeated it two more times. In the end, if the weevil failed to respond to all three artificial stimulations, the duration of TI was recorded as 0.1 s.

### 2.3. Artificial Selection

Due to the individuals of the SPW showing a different strength in TI, the two-way artificial selection was conducted based on the duration of TI [27,28,29]. In two-way artificial selection, fifty males and fifty females of the SPW were chosen at random from the original culture and the duration of TI was measured. A selection was conducted to find ten males and females with the shortest TI duration as S-strain. On the other hand, ten males and ten females with the longest duration to train as L-strain. They were introduced to mate, oviposition, and feed on sweet potato tuber inside a plastic container for a week. The weevils in containers were removed. Then, the offspring of F0 and S0 were collected in Petri dishes. We randomly collected and measured the duration of TI of fifty males and fifty females from each strain when they emerged (14–16 days old). In L-strain, ten male and female subjects with the longest TI were chosen to reproduce L-2. In S-train, ten males and ten females with the shortest TI duration were chosen to reproduce S-2. For each strain, we repeated the procedure for ten generations. We randomly measured the time of TI of male and female weevils from L-10 and S-10. Both male (mean = 1016.32 ± 59.18 s n = 20) and female (1026.20 ± 52.30 s n = 20) weevils from L-strain showed a significantly longer duration of TI than male (mean = 76.57 ± 21.90 s n = 20) and female (69.43 ± 23.16 s n = 20) weevils from S-strain (Figure 2).

#### 2.3.1. Experiment 1: Testing Effects of Courtship and Copulation on the Duration of TI

This trial consisted of three treatments: courtship (n = 25), in which a male was courting a female; copulation (n = 25), in which pairs were engaged in copulation; and resting (n = 25), in which both male and female were motionless. Each treatment was conducted in both L-strain and S-strain. Small plastic dishes (5 cm diameter × 2 cm height) were used. The 14–16-day-old adult virgin males and females were weighed using an electronic balance and isolated for 12 h. During the isolation culture, the SPWs were fed on the sweet potato tubers under the following conditions: 25 ± 1 °C, 70–80% relative humidity, and a photoperiod of 12:12 h (light from 06:00 to 17:59). To increase the chance of courtship and copulation between male and female adults, the experiment was conducted from 21:00 to 2:59.

To prevent the weevil pairs from copulating, we used adhesive plaster (Band-Aid Water Block, Warren, PA, USA) to cover the genitalia of male and female adults before weighing and insolation. Then, a virgin pair with adhesive plaster were introduced into the same Petri dish containing 1 g of sweet potato tuber. When the male crawls on the back of the female for 10 min, both are artificially separated and induced into TI. The duration of TI of males and females was recorded. If the male did not mount within 60 min, the male and female were excluded from the data.

#### 2.3.2. Experiment 2: Testing Effects of TI on Courtship and Copulation

All SPWs were 14–16-day-old virgins in experiment 2. Before the observation of courtship and copulation behavior, male and female weevils were weighed and isolated as described above. During the isolation culture, the sweet potato tubers were provided and the SPWs were reared under the following conditions: 25 ± 1 °C, 70–80% relative humidity, and a photoperiod of 12:12 h (light from 06:00 to 17:59). Under this experiment, the frequency of courtship and copulation, duration of courtship and copulation, and the insemination success rate of copulation were measured in four treatments (LM×LF, LM×SF, SM×LF, SM×SF: LM—L-strain male; SM—S-strain male; LF—L-strain female; SF—S-strain female). L-9 and S-9 generations of SPW were used in courtship treatment. The SPWs from L-10 and S-10 were selected to investigate the effect of TI on copulation. For full detail see our earlier publication [27].

The weevil pairs were introduced into the clear plastic dishes (5 cm diameter × 2 cm height). The mounting and genitalia-inserting behavior of pairs in 4 treatments were observed during courtship and copulation. The male mounting the female for 10 min is considered courtship. When male weevils fail to mount females within 60 min, we assume that courtship does not occur. In courtship, if the male SPW did not finish mounting within 90 min, the data were excluded. In copulation, if the pairs engaged in copulation did not separate within 120 min, the data were excluded. Each treatment was replicated 30 times. In courtship, we randomly selected ten SPWs and measured the duration of TI and mounting in four treatments. Similarly, ten SPWs were randomly chosen and we tested the period of TI and copulation in four treatments. 

Pairs of males and females were introduced into a clear Petri dish as in experiment 1 and allowed to copulate. If copulation happens in 2 h, we collected the females after copulation. The collected female SPWs were reared for 1–2 days. To confirm whether females were successfully fertilized, spermatheca of collected females were dissected using fine forceps and needles in isotonic sodium under a binocular microscope at 30 magnifications (Leica MZ 9.5, Wetzlar, Germany). A glass side consisting of a spermatheca and 0.3 mL of sterilized water was observed in an optical microscope (Leica DM/LS 020-518.500, Wetzlar, Germany) by 400 magnifications to verify the presence or absence of sperm. The number of females with sperm in spermatheca was calculated (Figure 3). 

## 3. Data Analyses

All data were analyzed with R 3.4.3 [30], and graphs were plotted using “ggplot” {ggplot2} [31]. We used generalized linear mixed models (GLMMs) to analyze all data using the lmer function in the lm4 package [32]. To compare the duration of TI of the SPW in courtship, copulation, and resting phase, we used the gamma distribution and log link function. The phases of the mating behavior of the SPW were fixed factors. Age, weight, and replication were random factors. To confirm the effect of TI on mounting and copulation reaction, we used the binomial distribution and logit link function. The strains were fixed factors. The age, weight, and replication were the random factors (where 1 = courtship or copulation successfully happened, 0 = failed to happen). We used the gamma distribution and log link function to compare the duration of courtship and copulation in the L-strain and the S-strain. The strains were fixed factors. Age, weight, and replication were random factors. To test the difference in insemination in two artificial strains, we used the binomial distribution and logit link function in GLMM. The strains were fixed factors. Age, weight, and replication were the random factor (where 1 = successful insemination, 0 = failed insemination). After each model was fitted, the significance of different stages and strains of the SPW was assessed by a likelihood ratio test between models, with and without the factor of interest, using C^2^ testing in “drop1{stats}” [33].

## 4. Results

There was a significant effect of mating behavior on the duration of TI in the L-strain (male: χ^2^ = 57.68, *p* < 0.01; female: χ^2^ = 102.75, *p* < 0.01, Figure 4A). The resting group had durations in males and females of about two times as long as the courtship and copulation group. The duration of TI of SPWs in courtship was significantly shorter than SPWs in copulation (male: χ^2^ = 12.05, *p* < 0.01; female: χ^2^ = 21.99, *p* < 0.01), and the duration of SPWs in the stage of copulation is significantly shorter than the resting group (male: χ^2^ = 39.89, *p* < 0.01, female: χ^2^ = 73.83, *p* < 0.01), Figure 4B. We found a similar situation in the S-strain.

In experiment 2, strains significantly affected the frequency (χ^2^ = 13.23, *p* < 0.01) and duration (χ^2^ = 175.15, *p* < 001) of courtship (as shown in Figure 5). There was no significant difference in the frequency of mounting between LF×LM and SF×LM (χ^2^ = 10.133, *p* = 0.72). The frequency of mounting of pairs from the treatment of LF×SM (χ^2^ = 5.35, *p* < 0.05) and SF×SM (χ^2^ = 6.00, *p* < 0.05) is significantly higher than LF×LM. Consequently, there was no significant difference in the duration of mounting between LF×LM and SF×LM (χ^2^ = 0.39, *p* = 0.53). The duration of mounting of pairs from the treatment of LF×SM (χ^2^ = 172.4, *p* < 0.01) and SF×SM (χ^2^ = 133.47, *p* < 0.01) was significantly shorter than LF×LM.

We found no significant difference in the frequency of copulation of pairs from four treatments (χ^2^ = 0.21, *p* = 0.97, Figure 6). However, the duration of copulation is affected by the strain (χ^2^ = 155.37, *p* < 0.01). There was no significant difference in the duration of copulation between LF×LM and LF×LMP (χ^2^ = 0.007, *p* = 0.93). The duration of copulation of pairs from the treatment of LF×SM (χ^2^ = 47.31, *p* < 0.01) and SF×SM (χ^2^ = 83.47, *p* < 0.01) was significantly shorter than LF×LM. In the comparison of pairs from different strains, we detected no significant difference in the insemination success rate (Figure 7, χ^2^ = 1.15, *p* = 0.76). We plotted the relationship between the duration of TI of males and courtship as well as copulation (Figure 8 and Figure 9). The male SPWs that showed a longer duration of TI had a longer duration of courtship and copulation. The duration of TI was significantly correlated with the duration of courtship (χ^2^ = 62. 36, *p* < 0.01) and copulation (χ^2^ = 44.89, *p* < 0.01).

## 5. Discussion

A trade-off was confirmed between TI and courtship and copulation. The SPWs showed a shorter duration of TI when they were courting or copulating than the SPWs which were motionless both in the L-strain and the S-strain. In the literature on the effects of TI on mating behavior, the male pairs from the S-strain showed a higher frequency of mounting than the males from L-strain. However, there was no effect of TI on the frequency of copulation. In further studies, we found that males of pairs derived from the S-strain had a significantly shorter duration of mounting and copulation than the males from the L-strain. However, there is no difference in the insemination success rate of the four kinds of pairs. This is growing evidence that a trade-off is occurring in anti-predator behavior and mating behavior in the SPW. Resource allocation conflicts happen when different activities increase the individual’s fitness and cannot be performed concurrently. In such situations, animals are predicted to increase fitness by trading off the benefit of one of the performance activities: courtship and copulation [34].

Prior studies have found that copulation reduces the duration of TI [26]. However, no study focuses on TI’s effect on courtship. In the present study, we measured the duration of TI of the SPWs from the L-strain and the S-strain during courtship and copulation. As per our hypothesis, the time of TI decreased during courtship and copulation. The cost of courtship and copulation has garnered significant interest. The price is considered to be involved in energy expenditures and enhanced predation risks [17,21]. It is also possible that the SPW moves during courtship and copulation, resulting in a shorter duration of TI. Prior studies have noted the importance of the effect of movement on TI in the SPW [5]. Furthermore, several reports have confirmed a negative association between mobility and TI.

The males of the SPW from L-strain pairs showed a longer duration of courtship and copulation than males from the S-strain. This suggests that male SPWs invest less in courtship and copulation. The neurotransmitter in the SPW may be explained by the reduction of courtship investment. Dopamine (DA) is crucial for controlling insect activity and is crucial for deciding mate success [35,36]. This is because DA is considered to be related to spontaneous locomotor activity in insects [37]. In previous studies, *Tribolium castaneum* with a longer duration of TI showed a higher level of DA than individuals with a shorter duration of TI. It is possible that the DA levels in the two artificial strains cause different expressions of mating behavior [38]. Other than that, the mobility discussed above is important for mating behavior [39]. Due to the negative association between mobility and TI, the SPW invests more time in mating.

There is no significant difference in the frequency of copulation and insemination success rate of the four kinds of pairs. This finding is consistent with that of Nakayama and Miyatake, who found TI affects mating behavior in *Tribolium castaneum* [29]. This suggested that it takes a longer duration for the L-strain to court and copulate to ensure the same frequency of copulation and insemination success rate as the S-strain. *T. castaneum* from a shorter duration of TI showed more copulation than the longer duration of TI individuals [29]. Thus, the SPWs from the L-strain need to prolong the time of copulation to increase the success rate of insemination. There is a positive correction between the duration of copulation and the percentage of eggs fertilized in *Pisaura mirabilis* [40]. The exact frequency of copulation in both may be the result of the duration of the courtship. The pairs in which males had a longer duration of TI showed a significantly more extended period of mounting. It suggested that male SPWs need a longer duration of copulation to improve the frequency of copulation. The successive male mating requires direct energy expenditure and nutrient provision to females via ejaculation [41]. Even if there is the same insemination success rate in two strains, the SPWs of the L-strain needs to copulate for a longer time to increase the mating success rate. To increase the success rate of mating, there are some costs in males performing longer duration of TI, such as longevity and immunity [42,43].

In summary, we confirmed that there is a trade-off between TI and courtship as well as copulation behavior. This gives us insight into how reproduction affects anti-predator behavior to increase the predation risk of prey. It provides support that there is a trade-off between mating and anti-predator behavior. The investment trade-off between TI and mating behavior offers technical support for the biological control of the SPW, for example, in order to improve the effectiveness of the use of the sterile insect technique in the SPW. The SPW with a shorter duration of TI should be used in the field because it is more efficient at mating. At the same time, understanding the trade-off between mating and anti-predator behavior is an important topic for evolution and biological control.

## 6. Conclusions

Our study provided clear evidence that the SPW reduced the investment in TI in the stage of courtship and copulation. Furthermore, L-strain males of the SPW spent more time in courtship and copulation to allow mating success than S-strain males. Thus, the trade-off investment of mating behavior and TI was confirmed. We further understand the trade-off between anti-predator behavior and mating behavior. By measuring the time of TI, courting, and copulation, we were able to quantify the investment in anti-predator behavior as well as mating behavior. We believe these methods could be useful to determine the cost–benefit between anti-predator behaviors in the prey and predator in other species too.

## Figures and Tables

**Figure 1 insects-14-00073-f001:**
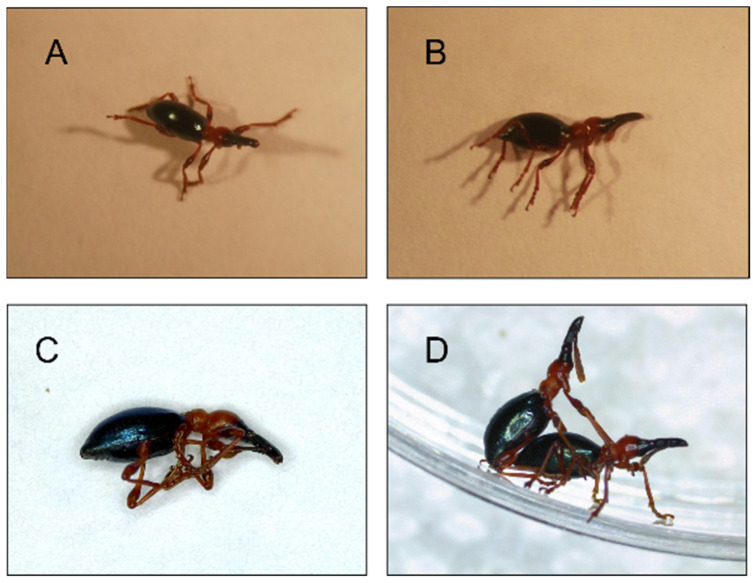
Adult sweetpotato weevil (*Cylas formicarius*): (**A**) resting sweetpotato weevil, (**B**) tonic immobility of the sweetpotato weevil, (**C**) dead sweetpotato weevil, (**D**) courtship in the sweetpotato weevil.

**Figure 2 insects-14-00073-f002:**
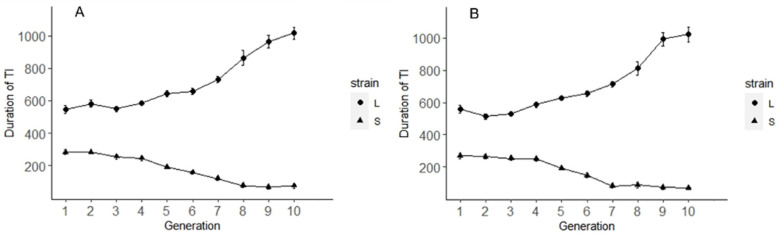
The duration of TI of SPWs from L-stain and S-strain. (**A**) The duration of TI of male SPWs from F1 to F10. (**B**) The duration of TI of female SPWs from F1 to F10. Circle—L-strain; triangle—S-strain.

**Figure 3 insects-14-00073-f003:**
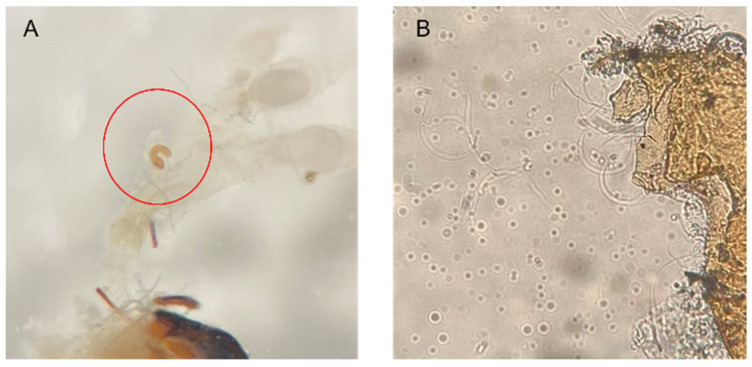
The sperm in spermatheca. (**A**) Spermatheca (shown in red circle) of females were dissected using fine forceps and observed with a microscope. (**B**) The observation of sperm in the spermatheca.

**Figure 4 insects-14-00073-f004:**
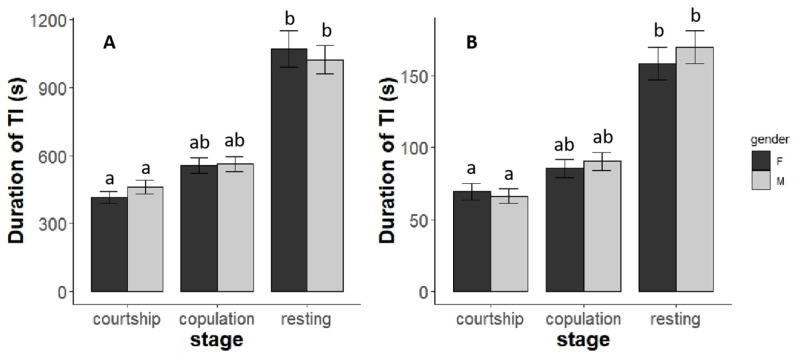
The duration of TI of 3 stages of female and male SPWs from L-strain and S-strain: (**A**) SPWs from L-strain. (**B**) SPWs from S-strain. Treatments with different letters are significantly different (*p* < 0.05) in a generalized linear model. Error Standard errors are shown on the column bars.

**Figure 5 insects-14-00073-f005:**
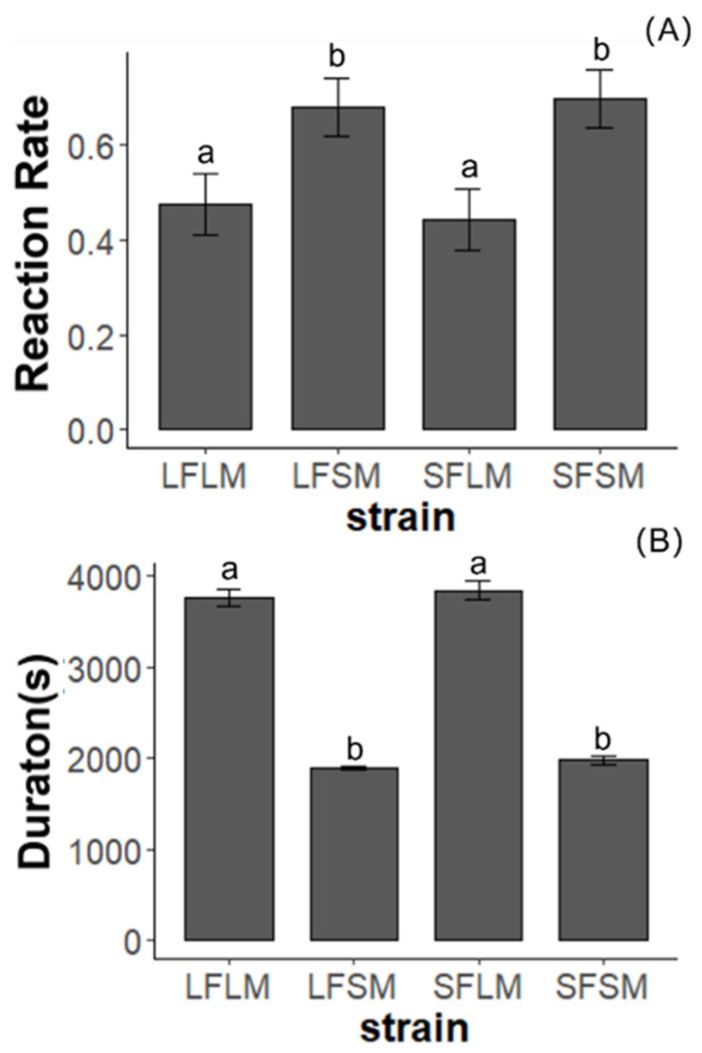
The courtship performance of the SPW for LF×LM, LF×SM, SF×LM, and SF×SM treatments. (**A**) The response of virgin males to females in four treatments. (**B**) The duration of courtship in four treatments. Treatments with different letters are significantly different (*p* < 0.05). Error Standard errors are shown on the column bars.

**Figure 6 insects-14-00073-f006:**
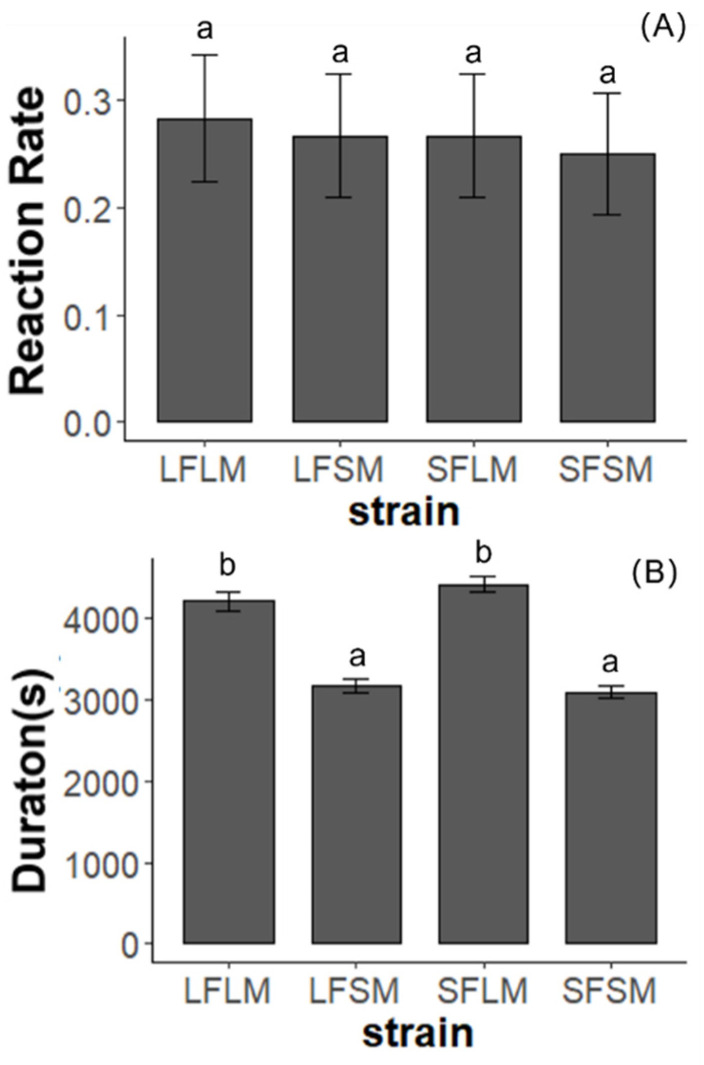
The copulation performance of the SPW for LF×LM, LF×SM, SF×LM, and SF×SM treatments. (**A**) The response of virgin males to females in four treatments. (**B**) The duration of copulation in four treatments. Treatments with different letters are significantly different (*p* < 0.05). Error Standard errors are shown on the column bars.

**Figure 7 insects-14-00073-f007:**
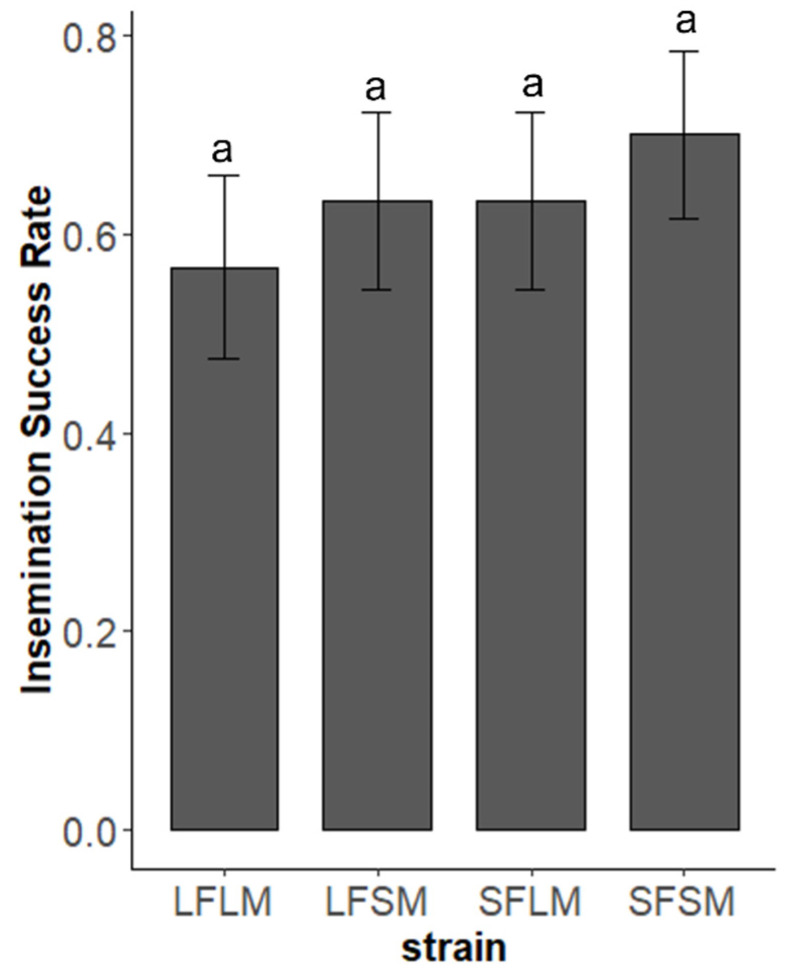
The insemination success rate of female SPWs after copulation in four treatments (LF×LM, LF×SM, SF×LM, SF×SM). Treatments with different letters are significantly different (*p* < 0.05) in a generalized linear model. Error Standard errors are shown on the column bars.

**Figure 8 insects-14-00073-f008:**
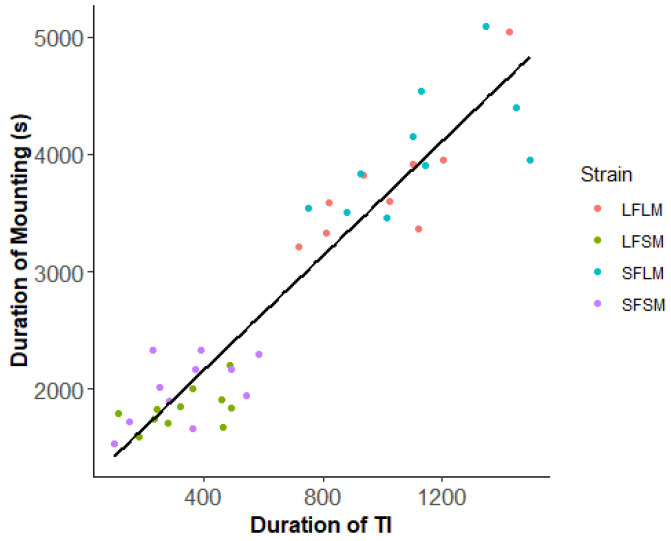
The relationship between the duration of TI and mounting in male SPWs. The four colors indicate different treatments: red—LF and LM; green—LF and SM; blue—SF and LM; purple—SF and SM (y = 1190 + 2.44T_ti_, R^2^ = 0.89).

**Figure 9 insects-14-00073-f009:**
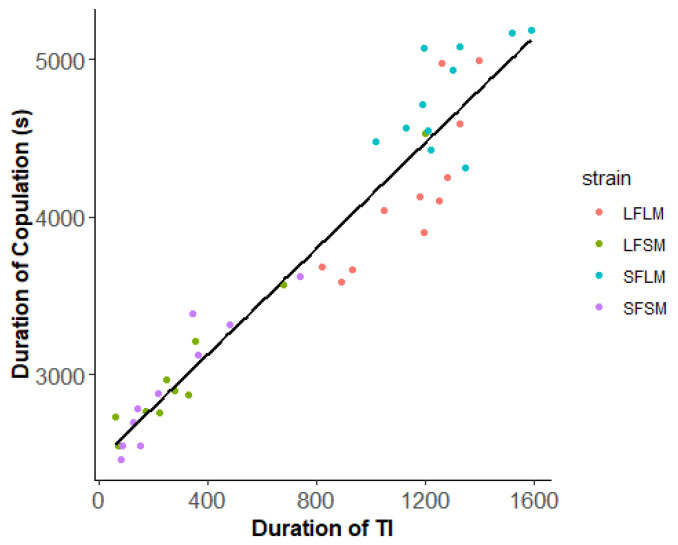
The relationship between the duration of TI and copulation in male SPWs. The four colors indicate different treatments: red—LF and LM; green—LF and SM; blue—SF and LM; purple—SF and SM (y = 2460 + 1.68T_ti_, R^2^ = 0.92).

## Data Availability

The data presented in this study are available on request from the corresponding authors.

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
