# Peer review of "Investment Trade-Off between Mating Behavior and Tonic Immobility in the Sweetpotato Weevil Cylas formicarius (Coleoptera: Brentidae)"

_insects, 2023, doi:10.3390/insects14010073_

Round 1

Reviewer 1 Report

Review for the manuscript ‘commentsInvestment Trade-off between Mating Behavior and Tonic Im- 2 mobility in the Sweetpotato Weevil Cylas formicarius (Coleop- 3 tera: Brentidae)’

Line 32: what are four kinds of pairs?

L 80: spell lab out

L 147: Show unit of duration

L 151: Write- were engaged in copulation

L 152: motionless or did not show any movement? -please express.

L 154: mention about reariing situation-fed/unfed, temp., humidity. Also, metion-how did you select these 25 males and female to ensure their normal health?

L 158: Did you confirm that this blocking of genitalia did not change their normal behavour? Please talk about that.

L 161: Due to the covering of genitalia, they made frequent efforts for copulation which probably caused them tired. As a result, the TI could not be the real expression-explain.

L 165: Age at which this action was performed. What does`=' mean here?

L 171: Please mention the sweet potato cultivar at the beginning of this study. Can  you mention the method of acclimatization?

L 176: Rewrite this sentence.

L 177: Write direct sentence.

L 178: Rewritej this sentence.

L 181: What is the purpose of selecting insects from L-9 and L-10 generations.

L 184: 21:00 - 24:00 EST?

L 185: Mention size of plastic dishes?

L 186: How did you confirm genitilia insertion? Did you observe it under microscope?

L 189: Did you observe whether it was not pretention?

L 193: You have to give some idea about how do you know exact time of male genitalia insertion?

L 195: This is not clear. This statement needs a time limit.

L 196: These statements are confusing. Please rewrite.

L 203: needs space between 2 and days.

L 208: Provide an image, if possible.

L 240: Use past tense (affected)

L 247: Change `is’ to `was’

L 248: Change `find’ to `found’

L 252: : Change `is’ to `was’

L 348: How can you come to this conclusion when you did not perform any predation experiment? This shows a biasness in drawing conclusion.

Review for the manuscript ‘commentsInvestment Trade-off between Mating Behavior and Tonic Im- 2 mobility in the Sweetpotato Weevil Cylas formicarius (Coleop- 3 tera: Brentidae)’

Line 32: what are four kinds of pairs?

L 80: spell lab out

L 147: Show unit of duration

L 151: Write- were engaged in copulation

L 152: motionless or did not show any movement? -please express.

L 154: mention about reariing situation-fed/unfed, temp., humidity. Also, metion-how did you select these 25 males and female to ensure their normal health?

L 158: Did you confirm that this blocking of genitalia did not change their normal behavour? Please talk about that.

L 161: Due to the covering of genitalia, they made frequent efforts for copulation which probably caused them tired. As a result, the TI could not be the real expression-explain.

L 165: Age at which this action was performed. What does`=' mean here?

L 171: Please mention the sweet potato cultivar at the beginning of this study. Can  you mention the method of acclimatization?

L 176: Rewrite this sentence.

L 177: Write direct sentence.

L 178: Rewritej this sentence.

L 181: What is the purpose of selecting insects from L-9 and L-10 generations.

L 184: 21:00 - 24:00 EST?

L 185: Mention size of plastic dishes?

L 186: How did you confirm genitilia insertion? Did you observe it under microscope?

L 189: Did you observe whether it was not pretention?

L 193: You have to give some idea about how do you know exact time of male genitalia insertion?

L 195: This is not clear. This statement needs a time limit.

L 196: These statements are confusing. Please rewrite.

L 203: needs space between 2 and days.

L 208: Provide an image, if possible.

L 240: Use past tense (affected)

L 247: Change `is’ to `was’

L 248: Change `find’ to `found’

L 252: : Change `is’ to `was’

L 348: How can you come to this conclusion when you did not perform any predation experiment? This shows a biasness in drawing conclusion.

Author Response

Review for the manuscript ‘comments Investment Trade-off between Mating Behavior and Tonic mobility in the Sweetpotato Weevil Cylas formicarius (Coleoptera: Brentidae)’

Line 32: what are four kinds of pairs?

Response: LF×LM, LF×SM, SF×LM, and SF×SM: male SPW from L-strain (LM), male SPW from S-strain (SM), female SPW from L-strain (LF), and female from S-strain (SF).

L 80: spell lab out

Response: The statement has been edited.

L 147: Show unit of duration

Response: The unit has been added.

L 151: Write-were engaged in copulation

Response: The missing words have been added.

L 152: motionless or did not show any movement? -please express.

Response: The resting SPW were motionless.

L 154: mention about rearing situation-fed/unfed, temp., humidity. Also, metion-how did you select these 25 males and female to ensure their normal health?

Response: The rear situation was described. To ensure their normal health, we observed whether the SPW eat sweetpotato tubers during isolation. If the SPW didn’t have any tubers in 12 hours, those were not used for further experimentation.

L 158: Did you confirm that this blocking of genitalia did not change their normal behavour? Please talk about that.

Response: We conducted pre-experiments to compare the response rate of mate searching and the frequency of courtship behavior in genitally blocked and normal males. The results revealed no significant difference between the two. This suggests that the use of adhesive plaster does not affect the mate-searching and courtship behavior of male SPWs.

L 161: Due to the covering of genitalia, they made frequent efforts for copulation which probably caused them tired. As a result, the TI could not be the real expression-explain.

Response: This is because the courtship time of male SPWs is generally between 2,000 and 4,000 seconds. The male can insert his genitalia into the female only after the female has accepted. In this study, the TI duration of the male was measured immediately after he had climbed on the back of the female for 10 minutes, so the results were not affected by the use of plaster.

L 165: Age at which this action was performed. What does`=' mean here?

Response: The 14-16 days old adult virgin males and females of the SPW were weighed by an electronic balance. Then, they were isolated for 12 hours and used in this experiment. As advised, The “=” has been removed.

L 171: Please mention the sweet potato cultivar at the beginning of this study. Can you mention the method of acclimatization?

Response: The cultivar of the sweetpotato was inserted. To ensure that the introduction of adults into a new environment does not have an effect on TI, after introducing the adult weevils into the Petri dish, let them get familiar with this environment for 30 minutes.

L 176: Rewrite this sentence.

Response: As advised, the sentence has been rewritten.

L 177: Write direct sentence.

Response: The sentence has been rewritten.

L 178: Rewrite this sentence.

Response: The sentence has been rewritten.

L 181: What is the purpose of selecting insects from L-9 and L-10 generations.

Response: The purpose was to illustrate the two experiments using different generations of the SPW separately. There were no longer significant changes in the duration of TI after the seventh generation. Therefore, there was no significant difference in TI behavior between the 9th and 10th generations.

L 184: 21:00 - 24:00 EST?

Response:  The EST has been added.

L 185: Mention size of plastic dishes?

Response: The size has been inserted.

L 186: How did you confirm genitalia insertion? Did you observe it under microscope?

Response: A microscope is not required to observe the copulation behavior of the SPW, the process of male genital insertion into female genitalia can be clearly observed with an aided eye.  

L 189: Did you observe whether it was not pretention?

Response: The process of copulation can be observed by unaided eyes. If the male weevil pretends to insert their genitalia into a female, pairs will separate within 5 minutes. Therefore, mating behavior was considered to occur only when the male inserted his genitalia into the female for more than 10 minutes.

L 193: You have to give some idea about how do you know exact time of male genitalia insertion?

Response: The process of male genital insertion into female genitalia can be clearly observed with the unaided eye, so we could measure the duration of copulation easily.

L 195: This is not clear. This statement needs a time limit.

Response: The statements were edited.

L 196: These statements are confusing. Please rewrite.

Response: As advised, the statements have been rewritten.

L 203: needs space between 2 and days.

Response:  The space has been added

L 208: Provide an image, if possible.

Response: The image has been added.

L 240: Use past tense (affected)

Response: Modified.

L 247: Change `is’ to `was’

Response: Modified.

L 248: Change `find’ to `found’

Response: Revised.

L 252: Change `is’ to `was’

Response: Revised.

L 348: How can you come to this conclusion when you did not perform any predation experiment? This shows a biasness in drawing conclusion.

Response: Although, no experiments on predators were conducted in this study, TI behavior serves as an important anti-predator behavior that can reduce the influence of predators on their prey. The relationship between TI and mating behavior was studied under this study, it can reflect the relationship between anti-predator behavior and mating behavior.

Reviewer 2 Report

Introduction

Authors must show some references to refer anti-predator behavior of insects.

Lines 130-147

For artificial selection, authors must write about selection intensity. It is also necessary to describe the duration of tonic immobility per generation, i.e. the selection response. Namely, authors should provide a figure or table showing generation and selection responses.

Line 137: “matting” should be “mating”.

Line 161: insolation ?

Lines 311-312: “In previous studies, 311 Tribolium castaneum with a longer duration of TI showed a higher level of dopamine (DA) 312 than individuals with a shorter duration of TI.” Cite the reference.

Author Response

Introduction 

Authors must show some references to refer anti-predator behavior of insects. 

Lines 130-147

For artificial selection, authors must write about selection intensity. It is also necessary to describe the duration of tonic immobility per generation, i.e. the selection response. Namely, authors should provide a figure or table showing generation and selection responses.

Response: The two-way artificial selection was conducted in artificial selection. The results of tonic immobility of SPW in each generation were recorded. In the article, I have shown the duration of TI of two strains of Sweetpotato weevils in the tenth generation. The figure or table showing the generation and selection responses was also inserted.

Figure: The duration of TI of the SPW from L-stain and S-strain. (A) Male SPW. (B.) Female SPW.

Line 137: “matting” should be “mating”.

Response: “matting” has replaced by “mating”

Line 161: insolation ?

Response: To prevent weevils from the same-sex sexual behavior and disturbance, each SPW used in the experiment was kept in separate plastic containers with 5g sweetpotato tubers for 24 hours.

Lines 311-312: “In previous studies, 311 Tribolium castaneum with a longer duration of TI showed a higher level of dopamine (DA) 312 than individuals with a shorter duration of TI.” Cite the reference.

Response:  As advised, the reference has been added.

Round 2

Reviewer 2 Report

The authors did not respond the following recommendation .

For artificial selection, authors must write about selection intensity. It is also necessary to describe the duration of tonic immobility per generation, i.e. the selection response. Namely, authors should provide a figure or table showing generation and selection responses.

Author Response

Thank you for the review. As advised, we have revised the manuscript and added the following:

Line 51:  In this approach, prey displays a distinctive posture that no longer responds to external stimuli when stimulated by predators or the environment".

Line 170 Figure 2 has been added.

Line 389:  We further know the trade-off between anti-predator behavior and mating behavior. 

Line 392:  By measuring the time of TI, courting, and copulation, we were able to quantify the investment in anti-predator behavior as well as mating behavior.
